# Persistence of Colistin Resistance and *mcr-1.1*-Positive *E. coli* in Poultry Despite Colistin Ban in Japan

**DOI:** 10.3390/antibiotics14040360

**Published:** 2025-04-01

**Authors:** Komei Kawano, Takayuki Masaki, Tatsuya Kawaguchi, Makoto Kuroda

**Affiliations:** Department of Medical Technology, Faculty of Health Sciences, Kumamoto Health Science University, 325 Izumi-machi, Kita-ku, Kumamoto 861-5598, Japanmasaki.ta@kumamoto-hsu.ac.jp (T.M.); kawaguti@kumamoto-hsu.ac.jp (T.K.)

**Keywords:** broiler, colistin, *E. coli*, *mcr-1*, IncI2, ST1485, zinc tolerance

## Abstract

Colistin, a polymyxin antibiotic used as a last-resort treatment for serious infections, exhibits efficacy against multidrug-resistant organisms. Colistin-resistant bacteria limit the treatment options and increase the risk of untreatable infections. In this study, we investigated various antimicrobial-resistant *Escherichia coli* strains isolated from broiler cecal feces. In the primary screening using CHROMagar C3GR and ESBL, 147 *E. coli* isolates were obtained from 231 broiler cecal samples at five domestic poultry farms in Japan in 2024. Of the 147 isolates tested for antimicrobial susceptibility, 20 (13.6%) showed resistance to colistin. Moreover, whole-genome sequencing detected the colistin resistance gene, *mcr-1.1* (phosphoethanolamine transferase), in the colistin-resistant *E. coli* strains isolated from the tested five poultry farms. Multilocus sequence typing revealed that all strains belonged to ST1485, indicating that the cloned strains had spread to multiple poultry farms. Subsequent core-genome comparison analysis with global ST1485 strains indicated that the ST1485 isolates in this study were highly identical, whereas the global strains were distinct. The complete genome sequence of BroCaecum-55 contained *mcr-1.1* in a 62,716 bp IncI2 replicon plasmid (pBroCa-55-p2). In conclusion, *mcr-1.1*-positive colistin-resistant *E. coli* strains, which are rarely reported in Japan, were isolated from Japanese broilers, indicating that colistin resistance persisted even after the ban on colistin use as a feed additive in Japan in 2018. Our findings highlight the need for continuous monitoring of colistin-resistant bacteria in livestock to reduce the transmission risk to humans.

## 1. Introduction

Increased spread of antimicrobial-resistant bacteria and associated infections is a serious issue worldwide. The United Nations World Health Organization (WHO) estimated 700,000 deaths worldwide due to antimicrobial resistance (AMR) in 2019. In the absence of proper control, this number is estimated to increase to 10 million by 2050, exceeding the number of cancer-related deaths.

AMR is a major issue impacting humans, livestock, the environment, and the food industry. Although initially developed against human infectious diseases, antimicrobials are also widely used for animals, including livestock (cattle, pigs, and chickens), aquatic animals (farmed fish), and pet animals (dogs and cats). In 2021, the total amount of antimicrobials used in Japan [1] was 1795 tons, with 800 tons used as animal antimicrobials and 226 tons used as antimicrobial feed additives. Although it enhances the livestock production safety, antimicrobial use also increases the prevalence of antimicrobial-resistant bacteria in livestock.

WHO proposed a Global Action Plan on AMR at its General Assembly in May 2015, requesting all member countries to develop their own action plans within two years [2]. Therefore, the Ministerial Conference on Countermeasures to Combat Internationally Threatening Infectious Diseases held by the Cabinet Office in Japan revealed the Action Plan for AMR Control 2016–2020 in April 2016. Subsequently, the Action Plan for AMR Control 2023–2027 [3] was published based on the five pillars of the WHO Global Action Plan on AMR: (1) public awareness and education, (2) trend surveillance and monitoring, (3) infection prevention and control, (4) proper use of antimicrobial agents, and (5) research and development and drug discovery. Recently, international collaboration was added as the sixth pillar to the plan. The action plan outlines strategies and specific initiatives for all six areas.

The Action Plan for AMR Control emphasizes the importance of AMR control using a one-health approach. The AMR Action Plan 2023–2027 clearly highlights the importance of promoting AMR countermeasures and implementing trend surveys on antimicrobial-resistant bacteria isolated from humans, animals, food, and the environment. The recent spread of antimicrobial-resistant bacteria has posed major public health concerns on food hygiene, particularly livestock hygiene, in humans. In Japan, the Japanese Veterinary Antimicrobial Resistance Monitoring system monitors the antimicrobial use in livestock and AMR of field strains, foodborne pathogenic bacteria, and indicator bacteria. Additionally, the Japan Nosocomial Infections Surveillance system monitors the trends in and use of antimicrobial-resistant bacteria.

Previously, third-generation cephalosporin antibiotic ceftiofur was used to prevent bacterial infections during the vaccination of fertilized broiler eggs [4]. However, it increased the prevalence of third-generation cephem-resistant bacteria. Subsequently, the poultry industry in Quebec, Canada voluntarily restricted the use of ceftiofur, leading to the decrease in the prevalence of resistant bacteria [4]. However, the prevalence of resistant bacteria increased again upon the reintroduction of ceftiofur, indicating a causal relationship between increased cephem use and prevalence of cephem-resistant bacteria. Owing to concerns regarding the increased prevalence of cephalosporin-resistant bacteria, the poultry industry voluntarily restricted the use of ceftiofur in Japan in 2010.

Colistin (CL) exhibits bactericidal activity against gram-negative rods, such as *Escherichia coli* and *Pseudomonas aeruginosa*, and is among the few treatment options available for multidrug-resistant bacteria [5]. It is a critically important antimicrobial in human medicine. It is also widely used in veterinary medicine to prevent and treat infectious diseases and promote growth. However, its use in livestock has been re-evaluated in many countries owing to the emergence of plasmid-mediated CL resistance gene (*mcr-1*)-carrying gram-negative bacteria [6]. *mcr-1* encodes a CL resistance protein by adding phosphoethanolamine to lipid A of lipopolysaccharide, thereby altering the polarity of the outer membrane and reducing its affinity for CL [6]. This gene was first reported as a novel factor for CL resistance in China in 2016 [6]. It is present on the plasmid and poses a transmission risk to various bacteria, including *Enterobacteriaceae* members [7]. The spread of this CL resistance gene to multidrug-resistant bacteria, such as multidrug-resistant *Pseudomonas aeruginosa* and *Acinetobacter* that do not respond to carbapenems, will make effective treatment challenging [8].

*mcr-1*-positive *Salmonella* and *E. coli* strains have been retrospectively identified in bovine mastitis and swine bacteremia cases in 2012 and 2013, respectively [9], by searching the whole-genome sequence data archives of Japan. *mcr-1*-positive *E. coli* have also been isolated from healthy food-producing animals [10], with approximately 10% positive detection of *mcr-1* in broilers in 2014 and increasing detection thereafter. Pathogenic swine isolates exhibited high *mcr-1* positivity (50%) in 2014 [11], and multiple *mcr* variants (*mcr-1*, *-3*, and *-5*) were detected in the diseased swine in Japan [12]. To date, CL resistance gene variants (*mcr-1*–*10*) have been mainly detected in *Enterobacteriaceae* members, such as *E. coli*, *Salmonella*, and *Klebsiella pneumoniae* [7]. In Japan, the *mcr-1*-positive *E. coli* clinical isolate, *E. coli* ST5702, was first detected in 2017 [13]. *K. pneumoniae* and *E. coli* clinical isolates have also been detected in other regions of Japan [14].

In Japan, the Food Safety Commission conducted a risk assessment in 2017 focusing on the prevalence of antimicrobial-resistant *E. coli* due to CL use and indicated the risk of CL resistance as “moderate”. Subsequently, the designation of CL as a feed additive was revoked and its use was prohibited by the Ministry of Agriculture, Forestry, and Fisheries based on the Risk Management Measures Guidelines for Antimicrobial Feed Additives on 1 July 2018 [15]. Additionally, “limited use as a second-line antimicrobial” risk management measure was proposed for CL as a veterinary antimicrobial based on the Risk Management Measures Guidelines for Veterinary Antimicrobials.

In this study, we examined the antimicrobial-resistant *E. coli* strains in the cecal stools of livestock, particularly broilers, and determined their transmission risk to humans.

## 2. Materials and Methods

### 2.1. Samples

According to the one-health approach, broiler cecal stools were selected for this study owing to the high antimicrobial use in livestock and ease of AMR analysis of the intestinal microbiota as cecal stools are small and less contaminated.

In total, 231 cecal stool samples of poultry from five poultry farms (A–E) collected by domestic meat hygiene laboratories in 2024 were included in this study. Specifically, 18, 67, 41, 61, and 44 fecal samples were obtained from poultry farms A, B, C, D, and E, respectively. The sample size depended on the shipping amount for each farm size.

### 2.2. Evaluation of Bacteria

#### 2.2.1. Isolation and Identification of *E. coli* Strains

Specific *E. coli* strains were isolated using CHROMagar C3GR (bioMérieux, Marcy-l’Étoile, France) and tested using CHROMagar ESBL (bioMérieux) to detect the extended-spectrum β-lactamase (ESBL)- and AmpC β-lactamase-producing bacteria. The cecum of the broilers was aseptically incised (approximately 0.5 cm), and cecal feces were collected using a sterile swab. The feces were inoculated into a screening medium and incubated at 35 ± 1 °C for 16–18 h. *E. coli* strains were isolated for further screening. Basically, one strain was isolated from one broiler fecal sample, and multiple strains were isolated from the same fecal sample in 18 cases. The strains were further identified using API20E (bioMérieux) for *Enterobacteriaceae* and other gram-negative rods, according to the manufacturer’s instructions.

#### 2.2.2. Antimicrobial Susceptibility

Next, minimum inhibitory concentrations (MICs) were determined using the broth microdilution method with the Clinical and Laboratory Standards Institute (CLSI)-compliant Dry Plate DP45 (Eiken Chemical Co., Ltd., Tokyo, Japan), according to the manufacturer’s instructions. Susceptible (S), intermediate (I), and resistant (R) strains were determined according to CLSI document M100-ED34 [16]. The following 2-fold serial broth dilutions of antimicrobials were tested in DP45: piperacillin (PIPC, 2–64 µg/mL), tazobactam (TAZ, 4 µg/mL constant)/PIPC (2–64 µg/mL), cefepime (0.5–16 µg/mL), ceftazidime (1–32 µg/mL), cefozopran (0.5–16 µg/mL), CL (1–4 µg/mL), fosfomycin (32–128 µg/mL), imipenem (0.5–16 µg/mL), meropenem (MEPM, 0.5–16 µg/mL), gentamicin (GM, 1–8 µg/mL), amikacin (AMK, 4–32 µg/mL), tobramycin (TOB, 1–8 µg/mL), minocycline (MINO, 1–8 µg/mL), levofloxacin (LVFX, 0.5–4 µg/mL), ciprofloxacin (CPFX, 0.25–2 µg/mL), doripenem (1–8 µg/mL), aztreonam (AZT, 2–16 µg/mL), and trimethoprim/sulfamethoxazole (TMP/SMX, 0.5/9.5–2/38 µg/mL).

#### 2.2.3. PCR Testing of ESBL Genes

All obtained *E. coli* strains were tested for ESBL genes by the multiplex PCR method using the Cica Geneus ESBL Genotype Detection Kit 2 (Kanto Kagaku, Tokyo, Japan) covering the detection as follows: CTX-M-1 group, CTX-M-2 group, CTX-M-8 group, CTX-M-9 group, CTX-M-25 group, CTX-M chimera, ESBL-type GES, TEM, SHV.

### 2.3. Molecular Biology Techniques

#### 2.3.1. DNA Extraction

DNA was extracted from *E. coli* strains for next-generation sequencing (NGS) analysis. *E. coli* suspensions were prepared in 1% TE sodium dodecyl sulfate and inactivated at 65 °C for 2 h. Then, the inactivated bacterial solution was used to purify the genomic DNA using the MinElute PCR Purification Kit (QIAGEN, Venlo, The Netherlands), according to the manufacturer’s instructions.

#### 2.3.2. NGS Analysis

To determine the whole-genome sequences of the isolates, NGS libraries were constructed using the QIASeq FX DNA library kit (QIAGEN). To remove the adapter dimers interfering with NGS decoding, a library with an insert length of 350 bp was recovered via 1% TAE agarose electrophoresis and gel recovery. The library was sequenced via 250-mer paired-end MiSeq using the MiSeq Reagent Kit v2 (Illumina, San Diego, CA, USA) for 500 cycles.

Oxford Nanopore Technologies that provide long reads for complete genome determination were used. A library was constructed using the Ligation Sequencing gDNA-Native Barcoding Kit 24 V14 (Oxford Nanopore Technologies, Oxford, UK), according to the manufacturer’s instructions. Then, the library was loaded onto flow cell R10.4.1 in MinION Mk1b and analyzed using MinKNOW v24.11.8 (Oxford Nanopore Technologies).

#### 2.3.3. Genome Informatics Analysis

CLC genome workbench v.24 was used for adaptor trimming of Illumina data and subsequent de novo assembly. Draft genome assembly was annotated using the DNA Fast Annotation and Structure Tool (https://dfast.ddbj.nig.ac.jp/, accessed on 15 November 2024) [17]. Then, hybrid assembly of the Illumina short reads and Nanopore long reads was performed using Unicycler v.0.4.8 [18]. AMR genes were detected using ResFinder v.4.6.0 (http://genepi.food.dtu.dk/resfinder, accessed on 15 November 2024) [19], and multilocus sequence typing (MLST; Achtman; https://pubmlst.org/multilocus-sequence-typing, accessed on 15 November 2024) was performed using the genome sequence. Sequence types (STs) were determined by comparing with the sequences of seven housekeeping genes (*adk*, *fumC*, *gyrB*, *icd*, *mdh*, *purA*, and *recA*) [20]. Pan-genome analysis was performed using Roary, which identifies the core and accessory genomes from the annotated data [21]. Core-genome analysis was performed using Parsnp v.1.7.4 (fast reference-based single nucleotide polymorphism [SNP] calling and core-genome alignment) to align the SNPs and core genomes from genome sequences [22]. A circular plasmid map was generated using Proksee (https://proksee.ca/, accessed on 8 January 2025) [23]. Additionally, comprehensive plasmid analysis was performed using the Plasmid Sequence Database (PLSDB) v2024_05_31_v2 (https://ccb-microbe.cs.uni-saarland.de/plsdb2025/, accessed on 20 January 2025) [24].

## 3. Results

### 3.1. Antimicrobial Susceptibility of the E. coli Isolates Obtained from Broiler Feces

To investigate third-generation cephalosporin-resistant *E. coli* strains in broilers, 231 broiler cecal samples were obtained from five domestic poultry farms (A, B, C, D, and E) in Japan in 2024. Using CHROMagar C3GR and ESBL, 147 isolates were obtained during primary screening. Notably, the 147 strains used in the analysis were isolated from individual broilers, but in 18 cases, multiple strains were isolated from a single individual. Specifically, 11 (61.1%) of 18 specimens, 30 (44.8%) of 67 specimens, 22 (53.7%) of 41 specimens, 28 (45.9%) of 61 specimens, and 32 (72.7%) of 44 specimens were obtained from the A, B, C, D, and E poultry farms, respectively.

Although the *E. coli* strains were obtained using CHROMagar C3GR and ESBL, MICs of the tested antimicrobials (Appendix A) revealed that all 147 isolates were susceptible to TAZ/PIP, cefozopran, AZT, imipenem, meropenem, and doripenem. AMR rates were as follows: PIPC (64 isolates; 43.5%), cefepime (one isolate; 0.7%), ceftazidime (nine isolates; 6.1%), CL (20 isolates; 13.6%), fosfomycin (one isolate; 0.7%), GM (47 isolates; 31.9%), AMK (one isolate; 0.7%), TOB (12 isolates; 8.2%), MINO (10 isolates; 6.8%), LVFX (19 isolates; 12.9%), CPFX (20 isolates; 13.6%), and TMP/SMX (67 isolates; 45.6%). Many strains were resistant to aminoglycoside antimicrobial agents, such as GM, and some multidrug-resistant strains were resistant to fluoroquinolone antimicrobial agents. Thirteen isolates (8.8%) were resistant to aminoglycosides and fluoroquinolones, and 19 isolates (12.9%) were resistant to CL. Unexpectedly, PCR detection of ESBL using a multiplex PCR kit suggested that all strains were negative for CTX-M groups and ESBL-type GES.

Strains were classified as S, I, and R strains based on the MIC breakpoint suggested by CLSI M100 ED34 [16]. Subsequently, susceptibility profiles of the 147 isolates were clustered via UPGMA (Figure 1). Based on their susceptibility profiles, the isolates were classified into the following five major clusters (clusters 1–5; Figure 1) with specific resistance patterns: cluster 1 (resistant to GM and TMP/SMX), cluster 2 (almost susceptible), cluster 3 (resistant to GM and TOB), cluster 4 (resistant to PIPC, CL, and ST), and cluster 5 (resistant to PIPC, GM, TOB, MINO, LVFX, CPFX, and TMP/SMX). CL R or I phenotype was observed only in cluster 4, indicating strains in this cluster showed notable AMR in the livestock.

To characterize the AMR genes in notable CL-resistant strains, we performed whole-genome sequencing of the strains in cluster 4, with at least one strain from each poultry farm A, B, C, D, and E, respectively. Additionally, two CAZ-resistant strains (BroCaecum-53-1-1 and BroCaecum-323) were included, and in total, seven strains were selected for whole-genome analysis (Table 1).

### 3.2. Whole-Genome Sequencing of Antimicrobial-Resistant E. coli Strains

Whole-genome sequencing revealed that seven strains belonged to ST1485 and were resistant to CL, harboring the CL resistance gene, *mcr-1.1* (seven of seven isolates: 100%; Table 1). Notably, all strains harboring *mcr-1.1* also carried *bla*_CMY-2_, *aph(6)-Id*, *aph(3″)-Ib*, *qnrS1*, *sul2*, and *dfrA14* (Table 1).

### 3.3. Comparative Genome Analysis of E. coli ST1485 Strains

As the CL-resistant strains belonged to ST1485 (Table 1), we performed comparative genome analysis using ST1485 *E. coli* isolated worldwide using the PubMLST database (https://pubmlst.org/bigsdb?db=pubmlst_escherichia_isolates. acessed on 15 November 2024). Core-genome SNP analysis was performed using Parsnp v.1.7.4 [22]. Subsequently, SNP network analysis was performed using population analysis with reticulate trees v.1.7 software [25].

We observed over nine thousand nucleotide variations between the Japanese (BroCaecum-xxx) and China-3468 strains; however, both types belonged to ST1485 and were *mcr-1.1*-positive strains (Figure 2A). Other global strains also showed clear differences from the Japanese strain (BroCaecum-xxx; Figure 2A), indicating that the *E. coli* ST1485 are not the key genome feature to show the CL resistance with *mcr-1.1*.

To evaluate the clonality of the Japanese strains (BroCaecum-xxx), SNP network analysis was performed using the seven *mcr-1.1*-positive CL-resistant *E. coli* strains isolated in this study. These seven isolates were clonal and classified into two clusters (cluster A: 17-2, 321, and 323; cluster B: 53-1-1, 55, 135, and 258), with 152 nucleotide variations (Figure 2B). Thus far, these clusters are unique to the *E. coli* ST1485 strain carrying *mcr-1.1*.

Similar to core-genome analysis, pan-genome analysis using Roary also indicated that the Japanese strain (BroCaecum-xxx) was different from the China-3468 strain due to the low similarity in their pan-genomes (Figure 3).

### 3.4. Complete Genome Sequence of the mcr-1.1-Positive E. coli BroCaecum-55 Strain

To reveal the horizontal gene acquisition mechanism of *mcr-1.1*, the complete genome sequence of BroCaecum-55 (*mcr-1.1*-positive *E. coli* strain) was determined via hybrid assembly of the Illumina short reads (250-mer PE; 870,640 reads) and Nanopore long reads (mean length: 4897 bp; 559,728 reads; Table 2). Its circular complete genome sequence indicated that BroCaecum-55 contained four plasmids: 176,133 bp, 62,716 bp, 5875 bp, and 3373 bp (Table 2). Notably, *mcr-1.1* was localized on the second largest plasmid with the IncI2 replicon (pBroCa-55-p2).

The circular plasmid map of pBroCa-55-p2 in Figure 4 shows *mcr-1.1* located at 18,135–19,760 bp. Zinc transporters involved in zinc tolerance were localized in the exogenous gene regions most likely acquired via horizontal gene transfer.

The plasmid homology search using PLSDB revealed that pBroCa-55-p2 shared the highest identity (99.5%) with the *E. coli* plasmids (pEC507_2 and pEC521_1) isolated from wild boars in Japan (Kagoshima and Iwate Prefectures, respectively) in 2012.

## 4. Discussion

This study demonstrated the significantly high isolation efficiency (13.6%; 20 of 147 *E. coli* isolates) of CL-resistant *E. coli* strains from broiler feces; however, CL-resistant strains have not been detected in broilers since July 2018 due to the restrictions on CL use as a feed additive in Japan [15]. According to the annual report of the Japanese Veterinary Antimicrobial Resistance Monitoring system of Japan, CL resistance rate in broilers was 3.3% in 2017, 0% in 2018 and 2019, 0.8% (one of 121 isolates) in 2020, and 0% in 2021 [26] (Appendix A), indicating no increase in CL resistance rates. Moreover, CL resistance rate of swine *E. coli* in Japan was 2.4% in 2017, 6.0% in 2018, 2.5% in 2019, 2.2% in 2020, and 2.0% in 2021 [26] (Appendix A). In Japan, *mcr-1* was detected in pigs before the 2019 study [27]. Notably, CL resistance and *mcr-1* positivity rates in this study are higher than those reported in previous studies. In China, CL withdrawal policy in agriculture has significantly reduced CL resistance in animals and humans [28,29,30]. Therefore, the specific conditions in Japan are possibly due to increased CL resistance, as observed in this study.

In this study, *mcr-1.1*-positive CL-resistant *E. coli* strains were isolated from five poultry farms. MLST (Atchman) revealed that all strains belonged to ST1485. Comparative core-genome analysis indicated that the Japanese *mcr-1.1*-positive CL-resistant strains were different from other global ST1485 strains by several thousand SNPs (Figure 2A). Moreover, only a few SNPs were observed among the Japanese strains, indicating that they were clonal domestically disseminating strains in Japan (Figure 2B) that were not recently introduced from other countries. A plasmid sequence search against PLSDB revealed notable nucleotide identity (99.5%) between the pBroCa-55-p2 and Japanese wild boar *E. coli* plasmids (pEC507_2 and pEC521_1), further confirming the persistence of *mcr-1*-positive *E. coli* strains in Japanese livestock and wild animals.

CL use as a feed additive has been prohibited by law in Japan since 2018 [15]. Despite this, CL resistance is prevalent possibly due to the presence of a resistance gene close to *mcr-1.1*, which is essential for bacterial survival, on the plasmid, with *mcr-1.1* consistently maintained via continued selection pressure. Here, we determined the complete genome sequence of BroCaecum-55 and found *mcr-1.1* and the zinc transporter gene in the IncI2 replicon plasmid (pBroCa-55-p2; Figure 4). Zinc products, such as zinc oxide, are used as antimicrobial feed additives to reduce the impact of CL withdrawal [31]. As zinc transporters are involved in zinc tolerance, *mcr-1.1* on the same plasmid is maintained via heavy metal selection, even when CL use is prohibited. In the veterinary field, Cu is used as a feed additive in swine at levels potentially leading to selection pressure on Enterobacterales. Arai et al. revealed that *Salmonella* 4,[5],12:i:-ST34 carrying copper tolerance genes shows high persistence and colonization of the intestine [32], leading to its recent dominance among *Salmonella* STs.

Except the CL resistance gene *mcr-1.1*, we could not isolate any ESBL-producing Enterobacterales in this study. In 2012, voluntary cessation of ceftiofur use by the Japanese poultry industry annually decreased the prevalence of cephalosporin-resistant *Salmonella* from 29.2% in 2012 to 10.5% in 2015 [33]. The cefotaxime resistance rate in broiler-derived *E. coli* further decreased annually from 4.7% in 2017 to 2.1% in 2021 [26] (Appendix A). Therefore, ESBL-producing bacteria are rarely detected in livestock. In contrast, the CPFX resistance rate has slightly increased annually from 12.0% in 2017 to 14.5% in 2021 [26] (Appendix A), warranting careful monitoring and control.

## 5. Conclusions

In this study, *mcr-1.1*-positive CL-resistant *E. coli* strains, which are rarely reported in Japan, were isolated from Japanese broilers, indicating their clonal distribution and prevalence in farms. Development of new antimicrobial agents has reduced the use of CL in clinical practice, thereby reducing the risk of CL resistance in humans. However, improper CL use may lead to a similar situation as that observed in 2015, with *mcr-1.1* complicating the control of multidrug-resistant organisms. The sample size from five poultry farms (A–E) was very small in this study, warranting further investigations with larger sample sizes from various farms. In addition, as limitations of this study, longitudinal data collection and investigation of environmental and feed-related factors are needed to accurately assess CL resistance in Japan. Overall, our findings highlight the importance of the continuous monitoring of CL-resistant bacteria in livestock to reduce the transmission risk to humans.

## Figures and Tables

**Figure 1 antibiotics-14-00360-f001:**
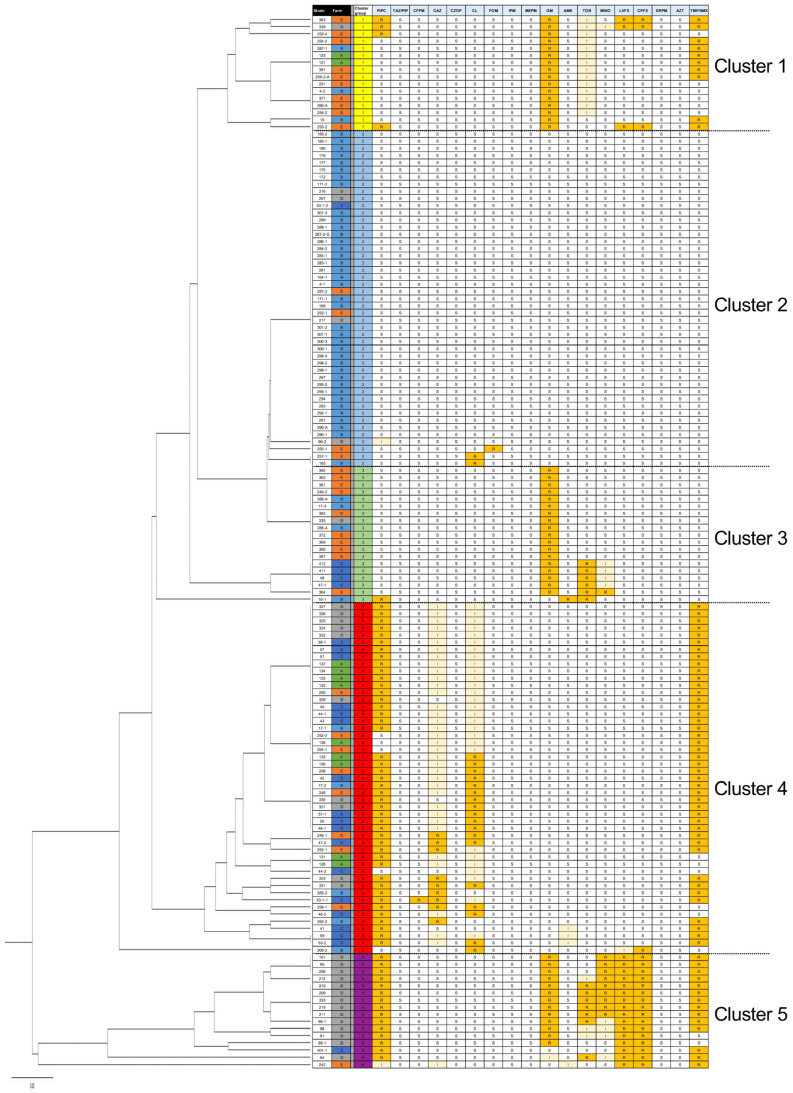
Unweighted pair group method with arithmetic mean (UPGMA) clustering of the antimicrobial susceptibility profiles of 147 broiler-derived *Escherichia coli* isolates. Antimicrobial susceptibility was determined using the broth microdilution method, and the obtained antimicrobial profiles (susceptible [S], intermediate [I], and resistant [R]) were clustered via UPGMA. R and I strains are highlighted by orange and light orange background, respectively. Strains in cluster 4 showed a notable colistin (CL)-resistant phenotype (Appendix A).

**Figure 2 antibiotics-14-00360-f002:**
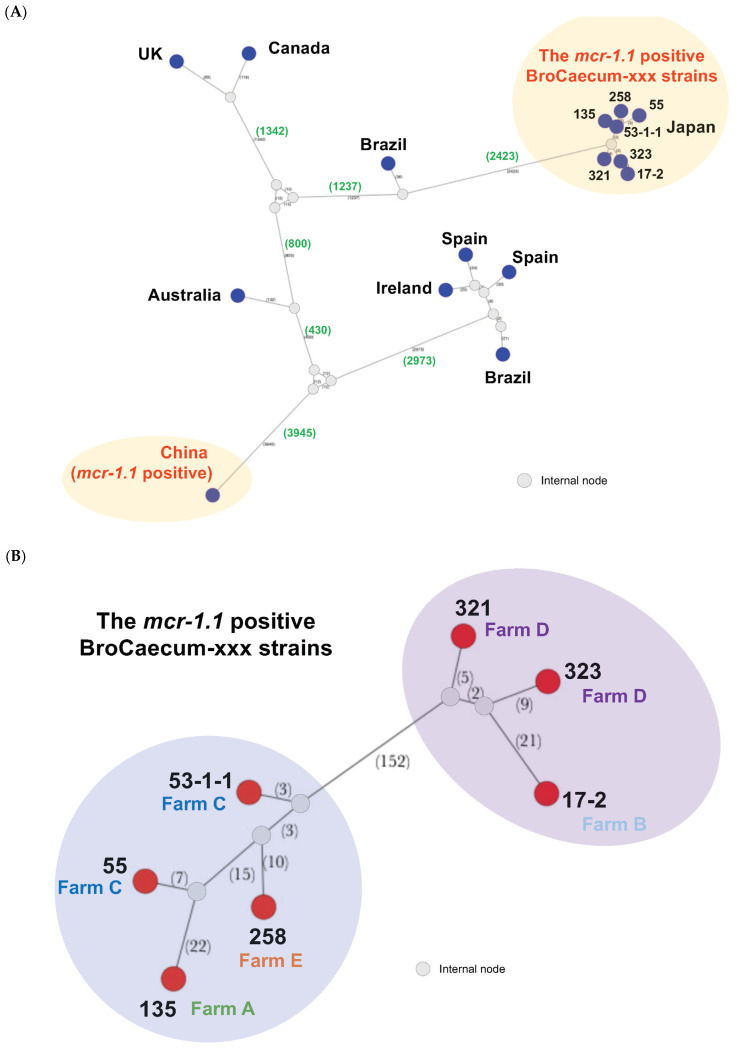
Core-genome single nucleotide polymorphism (SNP) network analysis of *E. coli* ST1485 isolates, including the *mcr-1.1*-positive colistin-resistant isolates. (**A**) Global ST1485 strains available from pubMLST database were included for the core-genome analysis. Seven Japanese isolates (BroCaecum-xxx) and one strain from China were *mcr-1.1*-positive. Dark blue nodes indicate the locations where the isolates were detected, and hypothetical internal nodes are shown in gray. Numbers of genomic SNPs are indicated in parentheses. (**B**) Subsequent core-genome analysis for seven Japanese isolates (BroCaecum-xxx). Red circle nodes indicate the strains, and hypothetical internal nodes are shown in gray. Numbers of genomic SNPs are indicated in parentheses.

**Figure 3 antibiotics-14-00360-f003:**
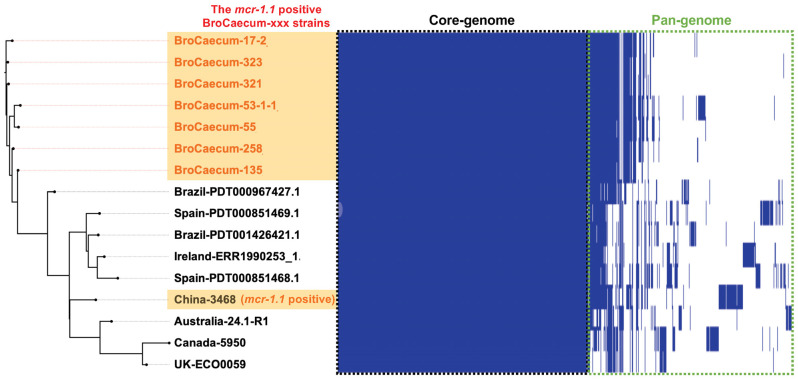
Pan-genome analysis of *E. coli* ST1485 strains using Roary. The *mcr-1.1*-positive BroCaecum-xxx strains were closely clustered in both pan-genome and core-genome SNP analyses (Figure 2B).

**Figure 4 antibiotics-14-00360-f004:**
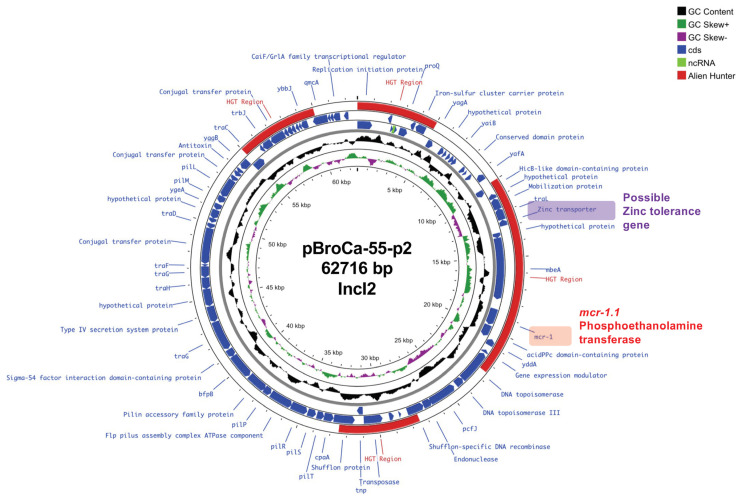
Circular map of the pBroCa-55-p2 plasmid from the BroCaecum-55 strain was analyzed using Proksee. Different circles indicate the scale (kbp), GC skew, GC content (average 42.8%), coding sequence (CDS) in forward orientation, CDS in reverse orientation, potential horizontal gene transfer (HGT) region predicted by the Alien Hunter program, and CDS annotation.

**Table 1 antibiotics-14-00360-t001:** Antimicrobial resistance genes in the genome sequences of *Escherichia coli* ST1485 strains identified in this study.

Strain	Farm	*aph(6)-Id*	*aph(3″)-Ib*	*bla* _TEM_	*bla* _CMY-2_	*mcr-1*	*qnrS1*	*tet*(A)	*sul2*	*dfrA14*
BroCaecum-17-2	B	+	+	+	+	+	+	+	+	+
BroCaecum-53-1-1	C	+	+	+	+	+	+	–	+	+
BroCaecum-55	C	+	+	+	+	+	+	–	+	+
BroCaecum-135	A	+	+	+	+	+	+	–	+	+
BroCaecum-258	E	+	+	+	+	+	+	–	+	+
BroCaecum-321	D	+	+	+	+	+	+	+	+	+
BroCaecum-323	D	+	+	+	+	+	+	–	+	+

+, positive; –, negative.

**Table 2 antibiotics-14-00360-t002:** Complete genome sequence of the *E. coli* BroCaecum-55 strain.

	Chromosome	pBroCa-55-p1	pBroCa-55-p2	pBroCa-55-p3	pBroCa-55-p4
Total length (bp)	5,098,687	176,133	62,716	5875	3373
Status	circular	circular	circular	circular	circular
Inc replicon	–	IncFIB	IncI2	rep_cluster_2401	–
Copy number	1.0	1.3	2.2	4.4	2.5
GC content (%)	50.6%	50.2%	42.8%	47.5%	55.2%
No. of CDSs	4648	181	74	7	4
No. of rRNA	22	–	–	–	–
No. of tRNA	86	–	–	–	–
MLST	ST1485	–	–	–	–
GenBank ID	AP039418	AP039419	AP039420	AP039421	AP039422

## Data Availability

All nucleotide sequence data analyzed in this study have been deposited into the DNA Data Bank of Japan (DDBJ) Sequence Read Archive under accession numbers PRJDB19974 and SAMD00874694–SAMD00874701. The complete and draft genome sequences have also been deposited into DDBJ and are listed in Table 2 and Appendix A.

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
