# Peer review of "Persistence of Colistin Resistance and mcr-1.1-Positive E. coli in Poultry Despite Colistin Ban in Japan"

_antibiotics, 2025, doi:10.3390/antibiotics14040360_

Round 1
Reviewer 1 Report
Comments and Suggestions for Authors
Peer Review Report_1
The manuscript presents an important investigation into the persistence of colistin resistance in E. coli isolates from poultry in Japan despite the colistin ban. The study is well-structured, and the findings provide valuable insights into AMR surveillance. However, several areas require clarification and improvement. Below are detailed comments.
Line 140: Please specify the concentrations of piperacillin (PIPC), tazobactam (TAZ)/PIPC, cefepime, ceftazidime, and other antibiotics tested in DP45.
Please justify the criteria for seven E. coli strains selected for WGS.
While the Discussion section presents key findings effectively, it could benefit from a more in-depth interpretation of the genomic comparisons, potential horizontal gene transfer mechanisms, and the broader implications of CL resistance persistence despite the ban.
The study includes a core-genome and pan-genome analysis, but the evolutionary implications of the observed genomic differences remain underexplored. Were any functional discrepancies identified between the Japanese E. coli strains and the global isolates?
The discussion mentions a fluctuating trend in CL resistance over the years but does not explain the observed variations clearly. Please add a brief discussion on possible confounding factors.
Figures 2 and 3 illustrate genomic relationships and should have clearer legends, such as indicating whether certain branches correspond to specific resistance genes or geographical origins.
I recommend major revisions before reconsideration for publication.
Author Response
We appreciate the reviewers’ helpful suggestions to improve the text of our manuscript. The following is a point-by-point response to the concerns of each reviewer. A word file with revision history was uploaded separately as a Non-published Material.
The initial draft presented 19 colistin-resistant strains, but in a simple mistake, we forgot to record one strain of BroCaecum-248.
In the revised manuscript, this has been corrected to 20 strains. In other words, we have corrected the frequency of colistin resistance to 20/147 (13.6%).
Reviewer 1:
The manuscript presents an important investigation into the persistence of colistin resistance in E. coli isolates from poultry in Japan despite the colistin ban. The study is well-structured, and the findings provide valuable insights into AMR surveillance. However, several areas require clarification and improvement. Below are detailed comments.
Line 140: Please specify the concentrations of piperacillin (PIPC), tazobactam (TAZ)/PIPC, cefepime, ceftazidime, and other antibiotics tested in DP45.
Reply to Reviewer:
We sincerely apologize for the lack of sufficient information regarding antimicrobial testing. The antimicrobial concentrations have been included in the revised Materials and Methods section.
Please justify the criteria for seven E. coli strains selected for WGS.
Reply to Reviewer:
Actually, we do not have sufficient funding to conduct whole-genome sequencing for all obtained strains. Therefore, we had to select representative strains to characterize the dissemination of the mcr-1.1 gene. This time, five strains from five farms and two CAZ-resistant strains were sequenced.
Please find the revised phrases below:
"We performed whole-genome sequencing of the strains in cluster 4, selecting at least one strain from each poultry farm (A, B, C, D, and E). Additionally, two CAZ-resistant strains (BroCaecum-53-1-1 and BroCaecum-323) were included, bringing the total to seven strains selected for whole-genome analysis (Table 1)."
While the Discussion section presents key findings effectively, it could benefit from a more in- depth interpretation of the genomic comparisons, potential horizontal gene transfer mechanisms, and the broader implications of CL resistance persistence despite the ban.
Reply to Reviewer:
We have avoided presenting overstatements using limited genomic comparisons. However, this study revealed that the persistence of Mcr-1.1 associated colistin resistance in clonal E. coli strains, which would be the main finding. We did not aim to overstate the manner of horizontal gene transfer.
The study includes a core-genome and pan-genome analysis, but the evolutionary implications of the observed genomic differences remain underexplored. Were any functional discrepancies identified between the Japanese E. coli strains and the global isolates?
Reply to Reviewer:
Indeed, we do not focus on the evolutionary aspects of colistin resistance, as mcr-1.1-positive E. coli strains appear to be a recent occurrence. These strains have disseminated globally since 2018. We speculated that core- (or pan-) SNP genome analysis, considering a few mutations among strains, would be the best approach to determine whether colistin resistance in Japan results from recent dissemination or persistent circulation.
The discussion mentions a fluctuating trend in CL resistance over the years but does not explain the observed variations clearly. Please add a brief discussion on possible confounding factors.
Reply to Reviewer:
Regarding the annual variation in colistin resistance frequency, the resistance rate has been very low since 2018, and I can't think of any special confounding factors. We are unable to add to the discussion as you have requested.
Figures 2 and 3 illustrate genomic relationships and should have clearer legends, such as indicating whether certain branches correspond to specific resistance genes or geographical origins.
Reply to Reviewer:
In the revised version RE1, Figures 2 and 3 were combined into Figures 2A and 2B. Here, we primarily demonstrated that Japanese mcr-1.1-positive E. coli strains are distinctly different from other global strains, including the mcr-1.1-positive strain from China. In particular, we found that Japanese E. coli ST1485 strains exhibit unique characteristics in carrying mcr-1.1 compared to other ST1485 strains. This finding suggests that the acquisition of mcr-1.1 may have occurred in Japanese poultry farms through plasmid transfer. However, we do not have concrete evidence to support this hypothesis.
Reviewer 2 Report
Comments and Suggestions for Authors
This study analyzed the antimicrobial resistance of 147 ESBL-producing E. coli strains against different types of antibiotics, with a primary focus on investigating the mechanism of colistin resistance. The content and results of the paper are valuable and have the potential to attract the interest and attention of researchers in the field. However, the logical flow of the study is somewhat unclear. Some key research findings are either missing or insufficiently presented, which weakens the overall impact and clarity of the study. Providing a more structured narrative and ensuring the inclusion of all critical results would enhance the coherence and scientific rigor of the paper.
Major comments:
- The primary focus of this study appears to be the exploration of colistin resistance in ESBL- or AmpC β-lactamase-producing E. coli isolated from various poultry farms in Japan. However, the main theme and logical flow of the article are somewhat unclear. Additionally, the authors do not provide a clear rationale for selecting ESBL- or AMPC β-lactamase-producing E. coli for analysis.
- Nineteen colistin-resistant E. coli strains were detected in this study, of which the authors selected seven for genome sequencing and identified all as ST1485. However, it is a pity that the MLST types of the remaining non-sequenced colistin-resistant E. coli strains were not determined.
- The authors performed whole-genome sequencing on seven strains but analyzed the sequence of only one strain, which is unclear and hard to understand. How about the other six strains? Do they harbor the same plasmids? Further clarification is needed.
- In Abstract, the description “147 cephalosporin-resistant E. coli 14
isolates were obtained…” is inaccurate, the author did not provide any evidence that they are cephalosporins resistant in this research. The MIC results indicated that only a few of them are resistant to the selected cephalosporins of this study.
- 5. The authors state in the discussion that no ESBL-producing coli were isolated in this study. However, this contradicts the overall logic of the paper, as the methods section clearly states that the 147 E. coli isolates were screened using an ESBL agar plate specifically for ESBL-producing bacteria.
Minor comments:
-There are only seven colistin resistant E.coli was selected for sequencing based on their source and antimicrobial resistance profile. Please clarify this in the abstract, it is confusing by saying “whole-genome sequencing detected…. E. coli strains isolated from all five poultry farms.”
-Line 147: What is “E.coli 22”?
-Line 195: Table S1 does not show the MICs but rather the antimicrobial susceptibility profiles of the 147 strains. Additionally, Table S1 is redundant with Fig. 1. Therefore, it is unnecessary to include Table S1 unless the actual MIC values are provided.
-Line 209-210: Please change to “: Cluster 1…., Cluster 2 …, Cluster 3…, Cluster 4 and Cluster 5…”
-Line 212: ‘indicating strains in this cluster…”
-Line 217: In Table 1, Is it “BroCaecum-53-1-1”? Because I did not see 53-1 in Cluster 4.
-Figure1, I am curious about the strains with an intermediate MIC of CL. Do they harbor any mcr genes?
-Line 255-256: I would prefer to name them as Cluster 1 and Cluster2. “These seven isolates were clonal and classified into Cluster 1 (17-2, 321, 255, and 323) and Cluster 2 (53-1-1, 55, 135, and 258), with 152 nucleotide variations.”
-Figure3, If strains 17-2, 321, 255, and 323 belong to the same cluster, they should be grouped within the same circular shading. The same applies to another cluster.
-Line 232:Please change “possessed” to “carried”
-Line 278: Please add ‘bp’ after each number.
Comments on the Quality of English Language
Moderate English language quality; requires minor revisions for clarity and flow.
Author Response
We appreciate the reviewers’ helpful suggestions to improve the text of our manuscript. The following is a point-by-point response to the concerns of each reviewer. A word file with revision history was uploaded separately as a Non-published Material.
The initial draft presented 19 colistin-resistant strains, but in a simple mistake, we forgot to record one strain of BroCaecum-248.
In the revised manuscript, this has been corrected to 20 strains. In other words, we have corrected the frequency of colistin resistance to 20/147 (13.6%).
Reviewer 2:
This study analyzed the antimicrobial resistance of 147 ESBL-producing E. coli strains against different types of antibiotics, with a primary focus on investigating the mechanism of colistin resistance. The content and results of the paper are valuable and have the potential to attract the interest and attention of researchers in the field. However, the logical flow of the study is somewhat unclear. Some key research findings are either missing or insufficiently presented, which weakens the overall impact and clarity of the study. Providing a more structured narrative and ensuring the inclusion of all critical results would enhance the coherence and scientific rigor of the paper.
Major comments:
- The primary focus of this study appears to be the exploration of colistin resistance in ESBL- or AmpC β-lactamase-producing E. coli isolated from various poultry farms in Japan. However, the main theme and logical flow of the article are somewhat unclear. Additionally, the authors do not provide a clear rationale for selecting ESBL- or AMPC β-lactamase-producing E. coli for analysis.
Reply to Reviewer:
Initially, as the primary focus of this study, we aimed to characterize the ESBL types in E. coli isolated from broiler feces. However, the obtained isolates did not carry ESBLs; instead, they were CMY-2 AmpC β-lactamase positive, as determined by partial PCR testing and whole-genome sequencing. After performing UPGMA clustering of these AMR profiles, we identified a unique CL-resistant cluster (cluster 4). Consequently, we shifted the main focus to CL resistance and its dissemination in E. coli.
- Nineteen colistin-resistant E. coli strains were detected in this study, of which the authors selected seven for genome sequencing and identified all as ST1485. However, it is a pity that the MLST types of the remaining non-sequenced colistin-resistant E. coli strains were not determined.
Reply to Reviewer:
Due to limited funding, we were unable to conduct whole-genome sequencing for all obtained strains. Therefore, we selected representative strains to characterize the dissemination of the mcr-1.1 gene. This time, five strains from five farms and two CAZ-resistant strains were sequenced.
Please find the revised phrases below:
“We performed whole-genome sequencing of the strains in cluster 4, with at least one strain from each poultry farm (A, B, C, D, and E). Additionally, two CAZ-resistant strains (BroCaecum-53-1-1 and BroCaecum-323) were included, bringing the total to seven strains selected for whole-genome analysis (Table 1).”
- The authors performed whole-genome sequencing on seven strains but analyzed the sequence of only one strain, which is unclear and hard to understand. How about the other six strains? Do they harbor the same plasmids? Further clarification is needed.
Reply to Reviewer:
As described above, we do not have sufficient funding to conduct Nanopore long-read sequencing for additional strains to determine their complete genome sequences. The core- and pan-genome analyses suggest that these seven strains are likely clonal. Therefore, one representative strain, BroCaecum-55, may be sufficient for determining the complete genome sequence.
- In Abstract, the description “147 cephalosporin-resistant E. coli
isolates were obtained...” is inaccurate, the author did not provide any evidence that they are cephalosporins resistant in this research. The MIC results indicated that only a few of them are resistant to the selected cephalosporins of this study.
Reply to Reviewer:
As the reviewer suggested, the description of cephalosporin resistance was incorrect. All MICs have been listed in the revised Table S2. Initially, the primary focus of this study was to characterize the ESBL types in E. coli isolated from broiler feces. However, the obtained isolates did not carry ESBL. Instead, some strains were positive for CMY-2 AmpC β-lactamase, as determined by partial testing using PCR and whole-genome sequencing. The description has been revised as follows: “using CHROMagar C3GR and ESBL, .......”.
- The authors state in the discussion that no ESBL-producing coli were isolated in this study. However, this contradicts the overall logic of the paper, as the methods section clearly states unless otherwise stated
that the 147 E. coli isolates were screened using an ESBL agar plate specifically for ESBL- producing bacteria.
Reply to Reviewer:
As described above, the relevant descriptions have been revised as follows: “using CHROMagar C3GR and ESBL, .......”.
Minor comments:
-There are only seven colistin resistant E. coli was selected for sequencing based on their source and antimicrobial resistance profile. Please clarify this in the abstract, it is confusing by saying “whole-genome sequencing detected.... E. coli strains isolated from all five poultry farms.”
Reply to Reviewer:
It is correct data that “mcr-1.1 positive E. coli strain was detected from the tested tested five poultry farms”. The description was a little revised to “the tested five poultry farms”.
-Line 147: What is “E.coli 22”?
Reply to Reviewer:
“22” was typo error, it was revised.
-Line 195: Table S1 does not show the MICs but rather the antimicrobial susceptibility profiles of the 147 strains. Additionally, Table S1 is redundant with Fig. 1. Therefore, it is unnecessary to include Table S1 unless the actual MIC values are provided.
Reply to Reviewer:
Table S1 has been revised to include the MICs. I would appreciate it if it met your request.
-Line 209-210: Please change to “: Cluster 1...., Cluster 2 ..., Cluster 3..., Cluster 4 and Cluster 5...”
Reply to Reviewer:
Thank you for the suggestion, it was revised.
-Line 212: ‘indicating strains in this cluster...”
Reply to Reviewer:
Thank you for the suggestion,
This has been revised, as suggested.
-Line 217: In Table 1, Is it “BroCaecum-53-1-1”? Because I did not see 53-1 in Cluster 4.
Reply to Reviewer:
Thank you for the suggestion,
It was revised as suggested to “53-1-1”.
-Figure1, I am curious about the strains with an intermediate MIC of CL. Do they harbor any mcr genes?
Reply to Reviewer:
Actually, draft genome sequence of all 147 strains should be determined, we do not have sufficient grants to conduct it for all obtained strains. Therefore, we have to select the representative strains. The strains in cluster 4 showed similar MIC profiles with CAZ resistance or intermediate, therefore, most strains might carry the mcr-1 gene.
-Line 255-256: I would prefer to name them as Cluster 1 and Cluster2. “These seven isolates were clonal and classified into Cluster 1 (17-2, 321, 255, and 323) and Cluster 2 (53-1-1, 55, 135, and 258), with 152 nucleotide variations.”
Reply to Reviewer:
The description was revised in the text, as suggested.
Figure 2B was revised, too.
-Figure3, If strains 17-2, 321, 255, and 323 belong to the same cluster, they should be grouped within the same circular shading. The same applies to another cluster.
Reply to Reviewer:
Figure 2B was revised.
-Line 232: Please change “possessed” to “carried” -Line 278: Please add ‘bp’ after each number.
Reply to Reviewer:
Thank you for the suggestion,
It was revised as suggested.
Reviewer 3 Report
Comments and Suggestions for Authors
The manuscript provides valuable insights into the persistence of colistin resistance and mcr-1.1-positive E. coli in poultry despite the colistin ban in Japan. This study is particularly interesting as it highlights the continued presence of colistin-resistant strains, raising concerns about the long-term impact of antimicrobial bans on resistance dynamics. However, I would like to seek clarification on a few aspects of the study:
Methodology: The study states that cecal stool samples were collected from five poultry farms (A–E), yet there is a noticeable discrepancy in the sample distribution across farms, with only 18 samples from Farm A, whereas Farm B had 67, Farm C had 41, Farm D had 61, and Farm E had 44. It would be beneficial if the authors could clarify the rationale behind this variation. Was it due to farm size, sampling constraints, or other factors?
Results: The study identifies seven mcr-1.1-positive colistin-resistant E. coli isolates, which were classified into two clones (BroCaecum-17-2, 321, 255, and 323; 53-1, 55, 135, and 258). However, only the BroCaecum-55 strain was selected for hybrid assembly to reveal the horizontal gene acquisition mechanism. I would appreciate further clarification on why this particular strain was chosen over the others. Were there specific genomic characteristics or epidemiological factors that made it the most suitable representative for this analysis?
Conclusion: The authors acknowledge small sample size as a study limitation, but I recommend providing a more comprehensive discussion of the study's limitations. For instance, additional aspects such as potential sampling bias, lack of longitudinal data, or the absence of environmental or feed-associated factors influencing colistin resistance could strengthen this section.
Addressing these points would enhance the clarity and robustness of the study's findings. I appreciate the authors' efforts in conducting this research and look forward to their response.
Author Response
We appreciate the reviewers’ helpful suggestions to improve the text of our manuscript. The following is a point-by-point response to the concerns of each reviewer. A word file with revision history was uploaded separately as a Non-published Material.
The initial draft presented 19 colistin-resistant strains, but in a simple mistake, we forgot to record one strain of BroCaecum-248.
In the revised manuscript, this has been corrected to 20 strains. In other words, we have corrected the frequency of colistin resistance to 20/147 (13.6%).
Reviewer 3:
The manuscript provides valuable insights into the persistence of colistin resistance and mcr- 1.1-positive E. coli in poultry despite the colistin ban in Japan. This study is particularly interesting as it highlights the continued presence of colistin-resistant strains, raising concerns about the long-term impact of antimicrobial bans on resistance dynamics. However, I would like to seek clarification on a few aspects of the study:
Methodology: The study states that cecal stool samples were collected from five poultry farms (A–E), yet there is a noticeable discrepancy in the sample distribution across farms, with only 18 samples from Farm A, whereas Farm B had 67, Farm C had 41, Farm D had 61, and Farm E had 44. It would be beneficial if the authors could clarify the rationale behind this variation. Was it due to farm size, sampling constraints, or other factors?
Reply to Reviewer:
The samples depended on the shipping amount for each farm size. This description has been included in the revised manuscript.
Results: The study identifies seven mcr-1.1-positive colistin-resistant E. coli isolates, which were classified into two clones (BroCaecum-17-2, 321, 255, and 323; 53-1, 55, 135, and 258). However, only the BroCaecum-55 strain was selected for hybrid assembly to reveal the horizontal gene acquisition mechanism. I would appreciate further clarification on why this particular strain was chosen over the others. Were there specific genomic characteristics or epidemiological factors that made it the most suitable representative for this analysis?
Reply to Reviewer:
As described above, we do not have sufficient funding to conduct Nanopore long-read sequencing for additional strains to determine their complete genome sequences. The core- and pan-genome analyses suggest that these seven strains are likely clonal. Therefore, one representative strain, BroCaecum-55, may be sufficient for determining the complete genome sequence. Indeed, there is no specific reason for selecting BroCaecum-55 in epidemiological factors.
Conclusion: The authors acknowledge small sample size as a study limitation, but I recommend providing a more comprehensive discussion of the study's limitations. For instance, additional aspects such as potential sampling bias, lack of longitudinal data, or the absence of environmental or feed-associated factors influencing colistin resistance could strengthen this section.
Reply to Reviewer:
Thank you for the valuable suggestion.
As the reviewer suggested, conclusions was revised.
Addressing these points would enhance the clarity and robustness of the study's findings. I appreciate the authors' efforts in conducting this research and look forward to their response.
Reply to Reviewer:
Thank you for the all of valuable suggestions.
Reviewer 4 Report
Comments and Suggestions for Authors
In the current manuscript, that authors isolated 147 cephalosporin-resistant E. coli strains and determined that 19 of them exhibited colistin resistance. WGS identified the mcr-1.1 genotype which confers the resistance. All strains analyzed were also attributed to sequence type 1485. Full genomic assembly for one of the strains located mcr-1.1 to a replicon plasmid. Overall, the study was well planned and executed, and the paper was well written. I have a few suggestions:
- Please explain a bit more on the rationale of starting off the screens with cephalosporin-resistant strains; this may help the flow of the paper.
- Line 215: please clarify on the criteria for selection of the 7 strains out of the 19 colistin-resistant strains, aside from being representative of distinct farm sources.
- Regarding the choice of BroCaecum-55 for comprehensive assembly: please explain as to why no other strains were analyzed in the same fashion; as it might be possible that genomic architecture could vary among those different strains.
- Related to #3, regarding the putative zinc transporter gene in the proximity of mcr-1.1 on the replicon plasmid: assuming such layout exists in all colistin-resistant strains of interest, it would be interesting to do an extra simple testing of ZincR to validate its function. In addition, perhaps suitable in a future study, it would be interesting to experimentally demonstrate the putative horizontal gene transfer in a conjugation experiment, underlining the “piggy-back” nature of mcr-1.1 on the zinc transporter gene in the same plasmid.
Minor:
- It might be helpful to consolidate Figs. 2 and 3, and possibly also with Fig. 4, to make the layout of the paper more streamlined.
- Line 147: please correct on “E. coli 22”.
Author Response
We appreciate the reviewers’ helpful suggestions to improve the text of our manuscript. The following is a point-by-point response to the concerns of each reviewer. A word file with revision history was uploaded separately as a Non-published Material.
The initial draft presented 19 colistin-resistant strains, but in a simple mistake, we forgot to record one strain of BroCaecum-248.
In the revised manuscript, this has been corrected to 20 strains. In other words, we have corrected the frequency of colistin resistance to 20/147 (13.6%).
Reviewer 4:
In the current manuscript, that authors isolated 147 cephalosporin-resistant E. coli strains and determined that 19 of them exhibited colistin resistance. WGS identified the mcr-1.1 genotype which confers the resistance. All strains analyzed were also attributed to sequence type 1485. Full genomic assembly for one of the strains located mcr-1.1 to a replicon plasmid. Overall, the study was well planned and executed, and the paper was well written. I have a few suggestions:
- Please explain a bit more on the rationale of starting off the screens with cephalosporin- resistant strains; this may help the flow of the paper.
Reply to Reviewer:
Initially, as the primary focus of this study, we aimed to characterize the ESBL types in E. coli isolated from broiler feces. However, the obtained isolates did not carry ESBLs; instead, they were CMY-2 AmpC β-lactamase positive, as determined by partial PCR testing and whole-genome sequencing. After performing UPGMA clustering of these AMR profiles, we identified a unique CL-resistant cluster (cluster 4). Consequently, we shifted the main focus to CL resistance and its dissemination in E. coli.
- Line 215: please clarify on the criteria for selection of the 7 strains out of the 19 colistin- resistant strains, aside from being representative of distinct farm sources.
Reply to Reviewer:
Actually, we do not have sufficient funding to conduct whole-genome sequencing for all obtained strains. Therefore, we had to select representative strains to characterize the dissemination of the mcr-1.1 gene. This time, five strains from five farms and two CAZ-resistant strains were sequenced.
Please find the revised phrases below:
"We performed whole-genome sequencing of the strains in cluster 4, selecting at least one strain from each poultry farm (A, B, C, D, and E). Additionally, two CAZ-resistant strains (BroCaecum-53-1-1 and BroCaecum-323) were included, bringing the total to seven strains selected for whole-genome analysis (Table 1)."
- Regarding the choice of BroCaecum-55 for comprehensive assembly: please explain as to why no other strains were analyzed in the same fashion; as it might be possible that genomic architecture could vary among those different strains.
Reply to Reviewer:
As described above, we do not have sufficient funding to conduct Nanopore long-read sequencing for additional strains to determine their complete genome sequences. The core- and pan-genome analyses suggest that these seven strains are likely clonal. Therefore, one representative strain, BroCaecum-55, may be sufficient for determining the complete genome sequence. Indeed, there is no specific reason for selecting BroCaecum-55 in epidemiological factors.
- Related to #3, regarding the putative zinc transporter gene in the proximity of mcr-1.1 on the replicon plasmid: assuming such layout exists in all colistin-resistant strains of interest, it would be interesting to do an extra simple testing of ZincR to validate its function. In addition, perhaps suitable in a future study, it would be interesting to experimentally demonstrate the putative horizontal gene transfer in a conjugation experiment, underlining the “piggy-back” nature of mcr-1.1 on the zinc transporter gene in the same plasmid.
Reply to Reviewer:
Thank you for your suggestion.
This study revealed the persistence of. Mcr-1.1 associated colistin resistance in E. coli, that would be the main finding. Next, we will demonstrate horizontal gene transfer of the mcr-1 gene without colistin selection. In addition, we did not aim to overstate the manner of horizontal gene transfer because there were insufficient experimental data obtained.
Minor:
- It might be helpful to consolidate Figs. 2 and 3, and possibly also with Fig. 4, to make the layout of the paper more streamlined.
Reply to Reviewer:
Thank you for the suggestion.
Figure 2 and 3 were consolidated to Figure 2A and 2B, as suggested.
- Line 147: please correct on “E. coli 22”.
Reply to Reviewer:
“22” was typo error, it was revised.
Round 2
Reviewer 1 Report
Comments and Suggestions for Authors
The revisions and recommendations provided have been adequately addressed in the resubmitted manuscript. The requested modifications have been implemented satisfactorily, and the manuscript is now suitable for publication in its current form.
Reviewer 2 Report
Comments and Suggestions for Authors
Thank you for carefully addressing my previous comments. The revised manuscript has significantly improved and now clearly resolves the concerns raised.